# Intercellular Communication in Airway Epithelial Cell Regeneration: Potential Roles of Connexins and Pannexins

**DOI:** 10.3390/ijms242216160

**Published:** 2023-11-10

**Authors:** Mehdi Badaoui, Marc Chanson

**Affiliations:** Department of Cell Physiology & Metabolism, Faculty of Medicine, University of Geneva, 1211 Geneva, Switzerland; mehdi.badaoui@unige.ch

**Keywords:** intercellular communication, connexins, pannexins, lungs, epithelial repair, epithelial signaling, epithelial differentiation

## Abstract

Connexins and pannexins are transmembrane proteins that can form direct (gap junctions) or indirect (connexons, pannexons) intercellular communication channels. By propagating ions, metabolites, sugars, nucleotides, miRNAs, and/or second messengers, they participate in a variety of physiological functions, such as tissue homeostasis and host defense. There is solid evidence supporting a role for intercellular signaling in various pulmonary inflammatory diseases where alteration of connexin/pannexin channel functional expression occurs, thus leading to abnormal intercellular communication pathways and contributing to pathophysiological aspects, such as innate immune defense and remodeling. The integrity of the airway epithelium, which is the first line of defense against invading microbes, is established and maintained by a repair mechanism that involves processes such as proliferation, migration, and differentiation. Here, we briefly summarize current knowledge on the contribution of connexins and pannexins to necessary processes of tissue repair and speculate on their possible involvement in the shaping of the airway epithelium integrity.

## 1. Introduction

Establishment of an apicobasal polarity is essential to produce a functional airway epithelium in terms of vectorial ion transport and innate immune defense. The lining of the normal human tracheobronchial tree is classically described as a pseudostratified, ciliated columnar epithelium with all epithelial cells attached to the basal membrane, whereas only basal cells do not reach the apical surface. Basal cells function as pluripotent stem cells capable of renewing the ciliated epithelium [1,2]. The other major cell types of the conducting airway epithelium include goblet cells, which produce mucus, multiciliated cells, and club cells [3]. Groups of cells often coordinate their movements during normal development, wound repair, and cancer invasion. These coordinated migrations require dynamic cytoskeleton and cell junction remodeling, which depends on the integration of intercellular signaling. As gap junctions provide metabolic and electrotonic pathways between cells, thus mediating direct communication between cellular nets, they contribute to epithelial tissue homeostasis [4]. The importance of intercellular signaling communication in healthy and respiratory diseases, including asthma, acute respiratory distress syndrome (ARDS), chronic obstructive pulmonary disease (COPD), chronic rhinosinusitis (CRS), cystic fibrosis (CF), and pulmonary artery hypertension (PAH), has been reviewed previously [5,6]. Surprisingly, there is limited knowledge on the role of gap junctions in the processes underlying airway epithelial cell regeneration. In this brief review, we discuss selected reports that may bring perspectives on the molecular mechanisms linking intercellular signaling communication to the generation of a polarized airway epithelium. More specifically, we summarize evidence for the role of connexins and pannexins on key biological processes, including proliferation, migration, adhesion, and differentiation, as reported on a variety of cell models, but with relevance for airway epithelial cell repair. We also address channel-independent functions of connexins and pannexins. The goal of this review is to propose working hypotheses for future research.

## 2. Intercellular Communication

Gap junctions are made of connexins that assemble to connexons (or hemichannels) in the membranes of contacting cells, which then dock to each other to form intercellular channels [7,8]. Undocked connexons may also form functional channels at the plasma membrane, sharing similar functions with pannexin channels (pannexons). Connexons and pannexons release nucleotides to the extracellular environment, thus contributing to indirect intercellular communication through autocrine or paracrine activation of purinergic receptors [7,9].

In humans, the connexin genes consist of 21 members, subdivided into subfamilies alpha, beta, gamma, delta and epsilon, respectively, encoding connexins (Cx) identified by their predicted molecular weight [10]. In contrast, the pannexin (Panx) family consists of three members. The correspondence between current gene names and proteins is given in Table 1. Connexins and pannexins share a common topology, with four transmembrane domains that span the plasma membrane, two highly conserved extracellular loops, and intracellular C- and N-terminal domains. Gap junctions, connexons, and pannexons channel activity is tightly regulated by phosphorylation, intracellular Ca^2+^ and pH, membrane potential, and protein–protein interaction, with specificities depending of the connexin or pannexin type expressed [4,11,12]. In addition, the lifecycle of connexins is dependent on endo-/lysosomal and autophagosomal pathways within the cell type studied [13]. Thus, connexin and pannexin expression and their assembly into functional channels (gap junctions, connexons, pannexons) are subjected to spatiotemporal dynamics with the stage of tissue differentiation. Not surprisingly, the pattern and profile of connexin/pannexin expression can change substantially during development and regeneration of the airway epithelium.

## 3. Connexin/Pannexin Expression in the Airway Epithelium

The structure and composition of the airway epithelium differs along the respiratory tract, thereby creating different regions with varying functions. The respiratory tract can be divided into two morphologically distinct regions: the conducting and the respiratory airways (the latter is defined by the absence of cartilage in the airway wall). The conducting airway includes the nasal cavity, the pharynx, the larynx, the trachea, and the bronchi. Recent progress in high-throughput techniques for single-cell genomic and transcriptomic analysis has further underlined the complexity of the cell composition of the airway epithelium. Mostly constituted of basal, secretory, and multiciliated cells, these cell types can be genetically subdivided into different populations [14,15]. Submucosal glands and surface airway epithelial cells contribute to the secretion of antimicrobial molecules, including β-defensins and cathelicidins [16]. Crucially, particles and microbes are removed from the epithelium surface by a mechanism called mucociliary clearance. They are first trapped by a mucin layer, made of mucin 5B (MUC5B) and MUC5AC, being produced by submucosal glands and goblet cells, on top of an aqueous periciliary liquid [17]. Then, the mucous gel is moved up the airways by the beating of the motile cilia of multiciliated epithelial cells and eliminated by sputum expectoration or swallowing. Efficient mucociliary clearance is achieved by the continuous regulation of airway surface hydration by modulation of the cystic fibrosis transmembrane conductance regulator (CFTR) activity [18]. CFTR is a chloride and bicarbonate ion channel, mutations of which are causing CF. CFTR is markedly expressed in ionocytes, a rare cell type recently identified within the surface epithelium [19,20], mostly present in club and goblet cells, and to a lesser extent in basal/suprabasal cells [21]. The primary culture of human airway epithelial cells is a model of choice as it most closely recapitulates the morphology and transcriptional profile of the native epithelia [22]. Typically, basal progenitor cells are isolated from biopsies, amplified in vitro, seeded on Transwell^®^ inserts, and grown at an air–liquid interface (ALI) for several weeks to trigger full differentiation towards a mucociliated pseudostratified epithelium [23,24].

In a transcriptomic analysis previously performed by our group on well-differentiated primary human airway epithelial cell cultures [25], variable amounts of mRNAs for Cx25, Cx26, Cx30, Cx30.3, Cx31, Cx31.1, Cx32, Cx37, and Cx43 were detected (Figure 1A) as well as for Panx1 and Panx2, but not Panx3 (Figure 2A). Among these isoforms, Cx26, Cx43, and Panx1 transcripts are the most abundant. Recent single-cell RNA-seq analysis revealed the relative levels of connexin and pannexin expression in the different cell types that constitute the airway epithelium. Integration of the data extracted from the atlas of the human healthy airways [14] and from the multi-institute consortium [26] shows different mRNA expression patterns across the airway epithelial cell types. Most connexin mRNAs are detected in basal and secretory cells; Cx25 mRNA is found in multiciliated cells and pulmonary ionocytes, while Cx37 mRNA seems to be restricted to the multiciliated cells (Figure 1B). Panx1 and Panx2 mRNAs are detected in all cell types, with a predominance for Panx2 in multiciliated cells (Figure 2B).

These transcriptional gene expression profiles were not always confirmed at the protein level. Older reports using available antibodies confirmed only the expression of some of these connexins in vitro. For instance, Cx26 and Cx31 are present in basal cells while Cx30 is detected in luminal cells (likely club and multiciliated cells), as reported in primary cultures [27]. Modest expression of Cx43 is found throughout the airway epithelium, which is strongly increased in some respiratory diseases, as reported in CRS sinonasal mucosa biopsies [28,29]. In addition, Cx43 mimetic peptides were found to reversibly inhibit calcium wave propagation in airway epithelial cells from rabbit tracheal explants [30]. The expression of Cx37 in airway epithelial cells of the conducting airways has, however, not been observed. Panx1 has been identified in primary cultures of differentiated human airway epithelial cells [31]. It is diffusely expressed in undifferentiated cells but its expression increases during differentiation, being detected apically in multiciliated cells [31]. Finally, Cx26, Cx32, and Panx1 are also found in submucosal gland cells [31].

## 4. Roles of Intercellular Communication in the Well-Polarized Airway Epithelium

As a first line of defense, the integrity of the airway epithelium is necessary to eliminate inhaled debris and microbes by several mechanisms, including mucociliary clearance and innate immunity [17]. Intercellular communication is critical for the cell-to-cell spread of calcium waves, which control ciliary beating and activate calcium-dependent membrane channels [32,33,34]. Calcium wave propagation may be achieved by the diffusion of inositol triphosphate (IP3) via gap junctions or through the release of ATP to the extracellular space via pannexons; ATP would in turn stimulate purinergic receptors to generate calcium signaling in surrounding multiciliated epithelial cells [35]. Recently, tuft cells (also termed cholinergic chemosensory cells or brush cells), a rare cell subtype of the airway epithelium, were identified as sensors of succinate, a metabolite that accumulates during infection, by secreting acetylcholine [36,37]. Acetylcholine activates muscarinic receptors, leading to the generation of intracellular calcium that could propagate through the airway epithelium via gap junctions [25]. Gap junctions have also been involved in transepithelial fluid transport and mucus hydration, two important parameters of airway surface liquid (ASL) homeostasis. For instance, gap junctional intercellular communication (GJIC) was found to contribute to CFTR activity and fluid secretion in airway epithelial cell lines in response to adenosine, prostaglandin E2 (PGE2), and protease-activated receptors [38]. Pannexons were also found to contribute to ASL volume regulation. Lipoxin A4 (LXA4) is known to stimulate ASL volume increase by modulating chloride and sodium transport in human airway epithelial cells [39]. Interestingly, ATP released from Panx1 channels triggered LXA4 production via activation of the purinergic receptor P2RY11 [40]. In fact, Panx1 KO mice exhibit ASL dehydration and deficient mucociliary clearance [41]. In airway epithelial cells from COPD patients and in cell models mimicking viral infection with dsRNA, the release of ATP via Panx1 channels and subsequent P2Y2R activation was found to stimulate the expression and release of MUC5AC in an autocrine manner [42]. Thus, intercellular communication may provide pathways for the coordinated regulation of mucociliary clearance in the conducting airways.

Gap junctions are also components of the innate immunity defense system, mediating the cell-to-cell spread of proinflammatory and proapoptotic signals depending on the activated-pathogen recognition receptors [43]. In airway epithelial cell lines, Cx43-made gap junctions transmit calcium-dependent signaling following the activation of Toll-like receptor (TLR) 2 to activate nuclear factor (NF)-κB and the epithelial secretion of interleukin (IL)-8, which recruits neutrophils to sites of pulmonary infection [44]. The survival/apoptosis balance was found to be dependent on the regulation of Cx43-mediated GJIC upon TLR5 activation by flagellin, an abundant structural protein of the bacterial flagellum [45].

Connexins and pannexins may also represent targets to restrict the severity of inflammation. Excessive ATP in ASL from patients with asthma is considered as an inflammatory signal. In this context, blocking Panx1 channels did prevent airway hyperreactivity in an asthmatic mouse model [46]. Medina and collaborators recently showed that Panx1-mediated intercellular communication between T regulatory and T effector lymphocytes leads to the breakdown of ATP to adenosine via ectonucleotidases, with adenosine exhibiting an anti-inflammatory effect in mice challenged intranasally with dust mite allergens [47]. Allergic airway inflammation may also result from mitochondria dysfunction, suggesting that transfer of healthy mitochondria to airway epithelial cells may have therapeutic benefits [48].

## 5. Speculative Roles of Intercellular Communication in the Regenerating Airway Epithelium

The airway epithelium is usually quiescent during homeostasis, with low rates of turnover, but has a tremendous capacity to regenerate after injuries induced by allergens, toxins, pathogens, and/or mechanical wounding [49]. The regeneration of the airway epithelium is complex, and includes epithelial wound repair and differentiation to reconstitute a fully differentiated and functional tissue [50]. It implies the orchestration of several processes, such as proliferation, migration, polarization, and final differentiation, which are regulated largely by cell–cell and cell–extracellular matrix interactions through epithelial receptors and signaling pathways [16]. Within a few hours of injury, basal cells re-enter the cell cycle, proliferate, and migrate to cover up the injured area in a matter of days. Once this is reached, progenitor cells exit the cell cycle, return to their original state, or progressively mature to differentiated cells within a couple of weeks [2]. It is also widely accepted that club cells can serve as intermediate progenitors to give rise to multiciliated and secretory cells [51]. Finally, repaired epithelial cells then differentiate to reestablish a well-polarized epithelium and restore its integrity [50]. The formation of junctional complexes between adjacent airway epithelial cells supports the epithelium structural integrity, thereby protecting against external agents. It includes the tight junctions and the adherens junctions [52,53]. Both tight (claudin-mediated) and adherens (cadherin/catenin-mediated) junctions form tight bonds between membranes at the apex of the cells, thus establishing an apicobasal polarity [54]. In addition to adherens and tight junctions, intercellular communication is crucial for proper maintenance of airway epithelial biology.

The gene expression profile was studied at different stages of repair of the human airway epithelium in response to circular wounding of primary cultures polarized at ALI [25]. Upon wounding, the mRNA expression for multiple connexins was transiently disrupted. Cx26, Cx30, Cx31, Cx31.1, and Panx1 are increased during the initial steps of repair, characterized by proliferation and migration of epithelial cells. Cx43 is specifically increased at time of wound closure. On the other hand, lower expression of Cx32, Cx37, and Panx2 mRNAs is observed during the early phases of repair, while Cx25 and Cx30.3 expression is stable (Figure 3A,B). Although these transient changes suggest a role for intercellular communication during airway epithelium regeneration, how connexins and pannexins regulate and coordinate the different steps of wound repair remains deeply unclear.

### 5.1. Connexins and Proliferation

The regulation of basal cell activation and proliferation is critical for their self-renewal and differentiation to secretory and multiciliated cell lineages. Connexins are expressed in adult stem cells, and their self-renewal capacities have been associated with gap junctions or pannexons function [55]. Upon wounding of well-differentiated primary cultures of human airway epithelial cells, Cx26 expression is dramatically increased in a cell subpopulation that expresses cytokeratin 14 [56], a marker of active basal cells [50]. Cx26 is first observed in the cytoplasm followed by localization at cell–cell contacts 24–48 h after injury, and then returned to basal levels with progression of the differentiation process [57]. Chandrasekhar and collaborators showed in cancer cells that cAMP diffuses between Cx26-connected cells to inhibit cell division, a mechanism consistent with the tumor suppressor role ascribed to this protein [57]. Accordingly, the antiproliferative effect of Cx26 was confirmed in human bronchial epithelial cells [56]. An inverse relationship between the Krüppel-like transcription factor 4 (KLF4) and Cx26 expression has been observed in keratinocytes from klf4-deficient mice whilst silencing Cx26 markedly decreased KLF4 mRNA levels in human airway epithelial cell lines [56,58]. KLF4 acts as a transcriptional activator of epithelial genes and as a repressor of mesenchymal genes; thus, its increased expression is required to regulate epithelial–mesenchymal transition (EMT). It is interesting to note that KLF4 is not induced in CF airway epithelial cells undergoing repair after wounding, whereas the regenerated CF airway epithelium exhibits features of EMT [59]. Cx26 may also function as a hemichannel; it has been described that airway epithelial cell proliferation is controlled by ATP release, P2RY11 receptors stimulation, and K_ATP_ potassium channel activation [40,60,61]. The hydrolysis of extracellular ATP to adenosine leads to the activation of A2B membrane receptors, triggering intracellular signaling that also enhances Cx26 hemichannel activity in small airway epithelial cells [62,63]. Thus, Cx26 may be at the interplay between hemichannel-mediated purinergic signaling and gap junctional communication during airway epithelial cell proliferation.

Cx43 and Cx37 were also shown to regulate cell proliferation in other contexts. For example, the overexpression of Cx43 promotes the proliferation of activated human embryonic stem cells through cyclin-dependent kinases [64]. Cx43-mediated GJIC was also shown to control glandular morphogenesis by influencing cell cycle entry and mitotic spindle orientation through PI3K–αPKC signaling [65]. In both studies, PI3K/Akt signaling was an important upstream pathway in the regulation of proliferation. Other recent data showed that Notch-dependent regulation of Cx37 induces endothelial cell cycle arrest by sequestrating activated-ERK, which in turn promotes the FoxO3a/p27 pathway [66]. FoxO3a, a direct target of Akt, functions as a powerful inhibitor of the cell cycle [67]. Of note, Notch signaling is also enhanced during wound repair and has been shown to be crucial in dictating cell differentiation into secretory lineages, whilst preventing differentiation into the multiciliated cells [68,69,70]. Because of the importance of Wnt, Notch, and PI3K/Akt signaling during airway epithelium regeneration, the increased expression of multiple connexins may fine-tune the proliferation/differentiation balance at an early step of the wound repair process.

### 5.2. Connexins, Adhesion, and Migration

Epithelial cell spreading and migration represent significant steps of wound repair. It relies on active cell volume regulation and well-coordinated cell–cell and cell–extracellular matrix (ECM) interactions. The multifaceted role of connexins supports their potential involvement in all these cellular processes [71]. For example, the inhibition of cell migration in repairing human airway epithelial cells led to increased Cx26 expression [56]. This result suggests that Cx26 expression is important during wound repair. Importantly, GJIC mediated by Cx26 has been shown to reduce adhesion and increase migration selectively in transfected HeLa cells [72]. Recently, Cx43-made gap junctions were found to mediate solute flow and regulate volume dynamics in mammary epithelial cell spheroids exposed to osmotic stresses [73]. The GJIC-mediated volume change was associated with increased cell stiffness through cytoskeleton reorganization, a response inhibited by blockers of gap junction channels [74]. In fibroblasts, overexpression of Cx43 led to a loss of contact inhibition of proliferation and decreased focal adhesions, events that are responsible for their higher motility [75].

Cell adhesion orientates the front-to-rear polarity in collectively migrating cells [76]. The high turnover of focal adhesion sites during cell migration involves the tight regulation of focal adhesion kinase (FAK). FAK clusters with multiple cytoskeletal proteins and with the main ECM integrin receptors at focal adhesion sites. Interestingly, the interaction of Cx26 and Cx43 with FAK has been reported [77,78], indicating an interconnection between the cell–ECM signaling and connexins. Importantly, connexins can also crosstalk with integrins as reported in different cell systems. A direct interaction between Cx43 and α5β1 integrin was reported in osteocytes, an interaction that is required for Cx43 hemichannel opening in response to mechanical stimulation [79]. A functional coupling between integrins and connexins can also occur. Indeed, Cx30 regulates β1 integrin redistribution in astrocytes by decreasing laminin secretion [80]. Consequently, the recruitment and activation of the small Rho-GTPase Cdc42 at the leading edge of migrating astrocytes was perturbed. Other examples supporting a role of connexins in migration come from the Cx43 knockout (Cx43 KO) mouse model, which dies at birth because of heart malformation. Knockout of Cx43 affected the directional migration of epicardial cells [81]. Indeed, a tubulin-binding domain was identified in the C-terminus of Cx43, with the Cx43–tubulin interaction being required for the directionality of cell migration [81]. Moreover, embryonic fibroblasts isolated from Cx43 KO mice confirmed the defective directionality of migration associated with misalignment of actin stress fibers at the front edge and microtubule destabilization. A reorientation failure of the Golgi apparatus and the microtubule-organizing center was also observed in these cells, a mechanism responsible for the abnormal cell polarity during wound closure [82]. A similar defect in directionality of migration was observed after truncation of the C-terminus of Cx43 in bone-marrow-derived dendritic cells [83]. The C-terminal isoleucine residue of Cx43 is critical for the binding to ZO-1, a major scaffold protein. Interestingly, disruption of the interaction between Cx43 and ZO-1 shifts directional migration of endothelial cells to a more linear movement that enhances the rate of wound healing [84]. Recently, a short region within the carboxyl tail of Cx43 was found to impair glioma stem cell migration and invasion through interaction with Src, PTEN and FAK [85]. Whether the latter effects can be ascribed to channel-independent influences of Cx43 needs further investigation. In fact, it is proposed that connexins could also act as adhesion molecules independently of their channel function [86].

### 5.3. Connexins and Epithelial Integrity

The barrier function of the terminally differentiated airway epithelium relies on stable cell–cell adhesion complexes, controlled cytoskeleton remodeling, and the establishment of an apicobasal polarity. Although data on the airway epithelium are lacking, functional and physical interaction between connexins and several actors important for the establishment of tissue integrity was reported in other cell models. For instance, intracellular calcium mobilization, calcium wave propagation, and oscillation involved hemichannels and GJIC to trigger actomyosin contractility and cytoskeleton remodeling in the endothelium [87,88,89,90]. However, the contribution of connexins in epithelial integrity appeared to be mostly channel-independent [91]. Indeed, studies showed that the C-terminus domain of Cx43 could physically interact with β catenin, a component of adherens junctions. Importantly, truncation of the PDZ-binding domain of Cx43 seems to perturb terminal differentiation of keratinocytes, leading to a defect of the epidermal barrier in Cx43-mutant mice without compromising the formation of gap junctions [92]. In hepatocytes, immunoprecipitation of tight junction proteins, i.e., occludin, claudin-1, and ZO-1, retrieved Cx32 but not Cx26, which lacks a PDZ-binding domain [93]. The interaction between connexins and multiple cytoskeleton-associated proteins such as drebrins, cingulin, α-actinin 4, and α- and β-tubulin has also been reported in different cell systems [94,95,96,97], clearly establishing an important relationship between gap junction proteins and the cytoskeleton. It is proposed that connexins may stabilize the junctions and the cytoskeleton dynamic. Inversely, actin remodeling is crucial for the successful membrane trafficking and function of the connexins. For instance, depletion of desmoglein 1 (Dsg1), a component of desmosomes, in keratinocytes reduces the localization of Cx43 at the plasma membrane and triggers its degradation; decreased Dsg1 and Cx43 were associated with a more fragile epidermal barrier [98]. On the same line, Go and collaborators showed that overexpression of Cx26 in Calu-3 airway epithelial cells is protective against the barrier-disruptive agent Ouabain [99]. In this context, occludin was also shown to bind to Cx26 via coiled-coil domain interaction [100]. Thus, induction of Cx26-mediated GJIC in proliferating basal cells may represent a means to not only to repress their proliferation but also to progressively promote the formation of a tight monolayer, which may serve as a platform for later differentiation of repairing airway epithelial cells.

### 5.4. Connexins, Differentiation and EMT

It has long been described in the air-conducting human and ferret airway epithelia that Cx26, Cx32, and Cx43 are present during development, and then almost disappear from well-differentiated adult airway epithelial cells [101,102]. This observation suggests that signals inhibit the expression of gap junction proteins once full differentiation is achieved and/or that a transient intermediate state associated with connexin expression is strongly repressed to terminate the differentiation process. EMT and its reverse process, mesenchymal-to-epithelial transition (MET), occur naturally during the developmental stage of organ and tissue formation [103], fine-tuning the regulation of cell differentiation and de-differentiation [104]. Importantly, the dysregulation of EMT and its impact on airway epithelial regeneration was observed in multiple chronic lung diseases associated with defective epithelial repair [105,106,107]. EMT and MET are also hallmarks of cancer metastasis [108]. EMT refers to loss of apicobasal polarity, reorganization of the actin cytoskeleton, cell–cell detachment, and acquisition of migratory properties. Although the molecular interplay between connexins, differentiation, and EMT/MET regulatory components is still not well described in the repairing airway epithelium, multiple studies highlighted these interactions in other physiological and pathological cellular contexts. TGF-β signaling plays an important role during EMT. Indeed, TGF-β triggers Smad-mediated signaling that activates EMT-associated transcription factors (EMTa-TFs), such as Snail and Twist1, and ultimately results in the expression of N-cadherin and loss of E-cadherin [109]. Importantly, a positive correlation was observed between E-cadherin and Cx26, Cx32, and Cx43 expression changes in human specimens of colorectal and non-small-cell lung cancers [110,111]. In addition, Cx43-made connexon assembly was facilitated by E-cadherin, while this process is inhibited during EMT by N-cadherin in rat liver epithelial cells [112]. In other studies, internalization of Cx43 and Cx26 and their intracellular accumulation in pulmonary epithelial cells were associated with EMT activation [113]. Conversely, Yang and collaborators showed that TGF-β1-dependent N-cadherin expression enhances Cx43 at the membrane by a direct interaction, leading to enhanced GJIC in primary murine osteoblasts [114]. A positive feedback loop may also exist since connexin hemichannel activity seems to be important for the initiation of EMT. Indeed, a recent study on adult retinal pigment epithelial cells showed that inhibition of ATP release with the Cx43 hemichannel inhibitor tonabersat reduced TGF-β2 release and ultimately attenuated EMT [115]. With these observations, we may speculate that terminal differentiation of airway epithelial cells would require inhibition of EMT-induced signals. In this respect, KLF4 may play a key integrative role by suppressing EMT and TGF-β signaling, as shown in nontransformed MCF-10A mammary epithelial cells [116], mouse intestinal epithelial cells [117], mouse corneal epithelial cells [118], and CFTR-expressing human bronchial airway epithelial cells [59]. Indeed, KLF4 ablation affected the expression of epithelial apicobasal polarity markers and Cdc42, as well as cytoskeletal actin organization in these cell systems [116,117,119]. In fact, KLF4 levels are generally elevated in differentiated tissues, including human airway epithelial cells [47], as compared to proliferating cells and cancer cells [120]. Whether KLF4-dependent suppression of EMT and TGF-β signaling would explain the overall decreased expression of connexins in well-differentiated airway epithelial cells remains to be demonstrated.

## 6. Speculative Roles of Pannexins in the Regenerating Airway Epithelium

Functional pannexons are essential for ATP release to the extracellular environment. The local ATP increase will then activate the ionotropic or the metabotropic purinergic receptors responsible for, respectively, Ca^2+^ entry or Ca^2+^ release from the intracellular stores [121]. Of note, the C-terminus of Panx1 has been shown to interact directly with F-actin, actin-related protein 3 (Arp3), and collapsin response mediator protein 2 (Crmp2), regulators of microtubule polymerization and stabilization in neural cells [122,123,124]. Thus, the crosstalk between pannexins, purinergic signaling, Ca^2+^ regulation, and cytoskeleton remodeling suggests a potential role of pannexons in several steps of the epithelium regeneration.

Multiple studies reported that pannexins positively or negatively modulate wound healing in various contexts [125,126,127]. Panx1-mediated ATP release activates purinergic P2 receptors, which are responsible for the proliferation of neural stem and progenitor cells [128]. The positive feedback loop between P2X7 receptors and Panx1 in dendritic cells was also shown to regulate cell migration speed [129]. This loop enhances Ca^2+^ entry and activates calmodulin kinase II (CaMKII) and cytoskeleton reorganization. The Panx1-dependent activation of P2 receptors was also shown to regulate skeletal muscle development and regeneration [130]. In this study, a link between Panx1 expression and genes involved in cell adhesion was also reported. Indeed, the silencing of Panx1 reduced the expression of multiple matrix metalloproteinases, which in turn remodeled the ECM and perturbed the cell–ECM interaction [130]. Another study described how ATP release by Panx1 localization at the leading edge of migrating T cells is required for FAK phosphorylation [131]. Panx1 expression may then influence lymphocytes’ polarization and migration in response to chemokines [131]. Furthermore, Noort and collaborators showed that different stages of embryo development are associated with specific subcellular localizations and glycosylation profiles of Panx1. Although knockout of Panx1 in human-induced pluripotent stem cells (iPSCs) did not affect the expression of genes regulating their morphology, survival, or pluripotency, the endodermal and mesodermal populations were over-represented after Panx1 silencing [132]. Thus, in addition to the regulation of proliferative signals, Panx1 expression in iPSCs may dictate cell fate decision.

A transient increase in Panx1 mRNA expression is observed during airway epithelium repair after wound injury (Figure 3B). Recently, Lucas and collaborators described how Panx1 is fundamental for partnering epithelial and immune cells for efficient lung regeneration [133]. In response to injury, a caspase-dependent activation of Panx1 occurs in damaged epithelial cells. Panx1 channel opening indeed promoted epithelial proliferation through the release of ATP and other factors, which in turn induced amphiregulin (AREG) expression by lung macrophages. Amphiregulin is a mitogen that belongs to the epidermal growth factor receptor (EGFR) ligands and has been associated with airway epithelium proliferation and regeneration [133,134,135]. Whether pannexons contribute to cell proliferation, adhesion, migration, or differentiation will depend on the cell type, the purinergic receptors expression pattern, and the cellular microenvironment. Among all the purinergic receptors, only P2X4, P2Y1, P2Y2, and P2Y6 mRNAs were strongly expressed in well-differentiated primary cultures of human airway epithelial cells (Figure 4). Interestingly, P2Y2 was shown to be essential for wound healing in corneal epithelial cells through the modulation of Ca^2+^ propagation and Src and ERK phosphorylation [136]. The cell metabolic status is also critical for pannexon-dependent ATP release. The repair process requires a high demand for energy, and ATP is known to accelerate tissue regeneration [137,138]. Because ATP release depends on the amount of ATP produced by the mitochondria, a localized accumulation of mitochondria can fuel Panx1-dependent purinergic signaling [139]. Panx1 can also be activated by mechanical stretching [140]. In the airway epithelium, the transduction of cell membrane stretching to Panx1-mediated ATP release is controlled by RhoA and Rho kinase-dependent myosin light chain phosphorylation [141]. Therefore, the mechanotransduction of the highly dynamic ECM remodeling during wound repair [142] may affect Panx1 conformation and activity [143].

## 7. Concluding Remarks

Synchronized cell proliferation, dynamic cell adhesion, collective cell migration, and coordinated interactions of epithelial cells represent fundamental processes for airway epithelium regeneration. Here, we summarized studies in several cell systems, supporting the view that connexins and pannexins may contribute to these fundamental processes. We also outlined knowledge gaps that need to be filled in relation to the airway epithelium. There is still a considerable lack of molecular evidence defining the precise role of intercellular communication to each step of airway epithelium regeneration. Figure 5 illustrates speculative functions of connexins and pannexins during the processes that lead to a mature airway epithelium. Determining which connexins and pannexins are expressed at the protein level during airway epithelium repair is one of the important challenges to address. This would require a strict experimental setup to control the method of cell culture (ALI vs. monolayers), the differentiation status (days at ALI), and the immune-detection of relatively low abundant proteins. Transcriptional and protein profiling of connexins and pannexins along the different regions of the upper and lower airways would also help in understanding their specific function along the respiratory tree. The development of new methods that combine in situ hybridization and immunofluorescence may represent helpful tools to monitor their spatial and temporal expression.

Another fundamental question is to discriminate between the channel-dependent or channel-independent role of connexins and pannexins in repairing airway epithelial cells. In this context, a particular focus must be placed on their localization, as connexins and pannexins can be found in intracellular compartments [122,144,145]. The intracellular relocalization of connexins and pannexins depends on the regulation of their trafficking and degradation, which are controlled by phosphorylation, acetylation, glycosylation, and ubiquitination [12,146]. Thus, additional studies on connexin and pannexin post-transcriptional modifications during the regeneration process would provide insights into the canonical and/or the noncanonical functions of connexins and pannexons. As proper EMT and MET are required for the regeneration of the airway epithelium, an improper EMT/MET balance could result in lung pathologies if aberrantly modulated [147]. The tight interplay between connexins and EMT may thus point to an essential role of intercellular communication in fine-tuning EMT during wound repair. To date, whether the spatiotemporal changes in connexins and pannexins expression upon wounding are a consequence of EMT or are needed for acquisition of proliferative and migratory properties by repairing airway epithelial cells, remain unclear. Altering intercellular communication in experimental models would be helpful but, unfortunately, drugs to modulate specifically connexin and pannexin channel activity are lacking. Gene silencing approaches may also be challenging as airway connexins/pannexins may compensate for each other, as observed, for example, in keratinocytes [125,148]. The ongoing development of connexin- and pannexin-targeting peptides may help to address this question [149,150,151]. As the well-differentiated airway epithelium accommodates with low connexin expression, any changes in their expression and/or regulation may alter the repair process. This, in turn, would impair the integrity and homeostasis of the airway epithelium, which could be manifested in respiratory diseases.

## Figures and Tables

**Figure 1 ijms-24-16160-f001:**
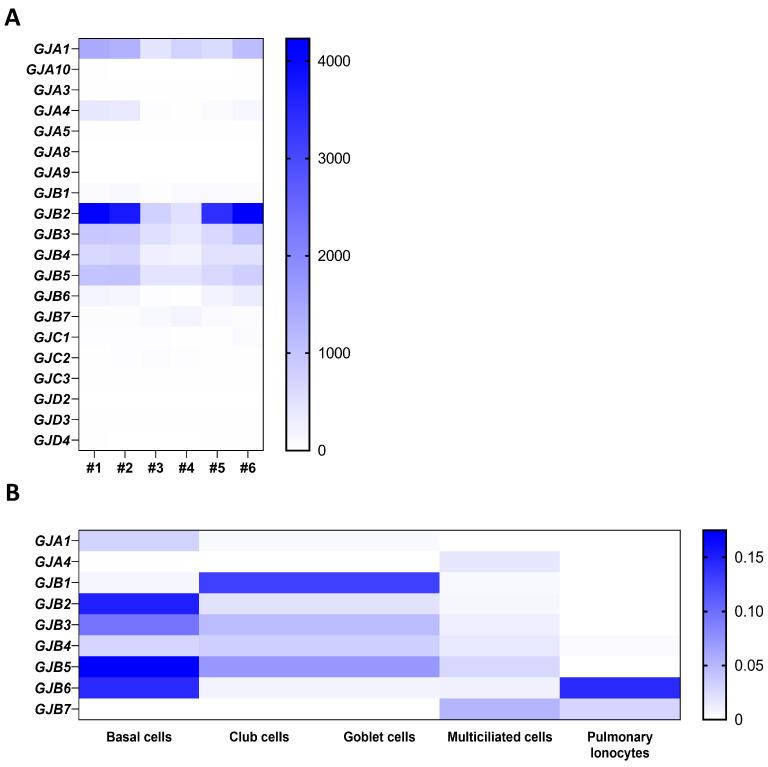
Gene expression of connexin isoforms in well-differentiated primary cultures of human airway epithelial cells. Heat map showing raw count data of connexin (**A**) isoforms obtained by RNA-seq on primary cultures of human airway epithelial cells in 6 individuals [25]. The source data can be accessed from the NCBI Gene Expression Omnibus (GEO) with the accession number GSE127696. (**B**) Relative expression of connexin mRNAs in different airway epithelial cell types, as evaluated by integrating single-cell transcriptomic data from the atlas of the human respiratory system [14] and the multi-institute consortium study [26] carried out on human airway tissues.

**Figure 2 ijms-24-16160-f002:**
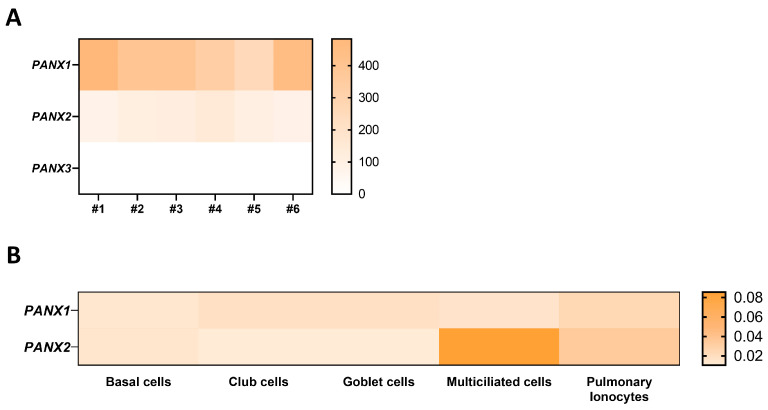
Gene expression of pannexin isoforms in well-differentiated primary cultures of human airway epithelial cells. Heat map showing raw count data of pannexin (**A**) isoforms obtained by RNA-seq on primary cultures of human airway epithelial cells in 6 individuals [25]. The source data can be accessed from the NCBI Gene Expression Omnibus (GEO) with the accession number GSE127696. (**B**) Relative expression of Panx1 and Panx2 mRNAs in different airway epithelial cell types, as evaluated by integrating single-cell transcriptomic data from the atlas of the human respiratory system [14] and the multi-institute consortium study [26] carried out on human airway tissues.

**Figure 3 ijms-24-16160-f003:**
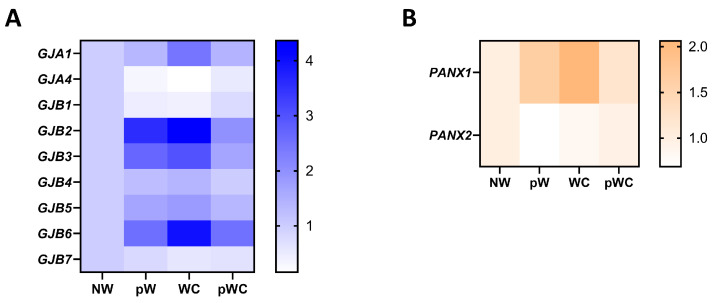
Gene expression profile of connexins and pannexin isoforms in repairing primary human airway epithelial cells. Fold-change heat map of detected connexin (**A**) and pannexin mRNAs (**B**) in primary human airway epithelial cells at different times of wound repair normalized to the nonwounded condition (NW). pW = 24 h post wound, WC = wound closure, pWC = 48 h post wound closure. The data are from [25] and can be accessed from the NCBI Gene Expression Omnibus (GEO) with the accession number GSE127696.

**Figure 4 ijms-24-16160-f004:**
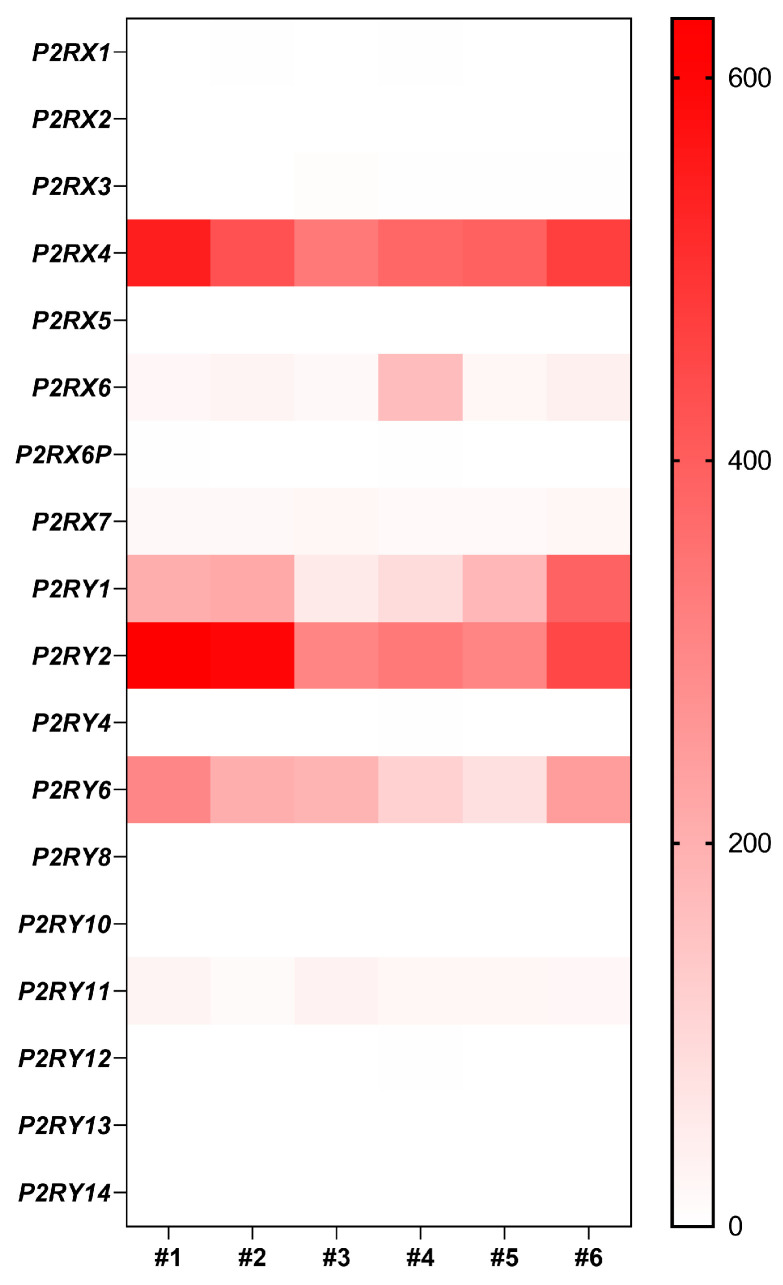
Gene expression of purinergic receptors in well-differentiated primary human airway epithelial cells. Heat map showing raw count data of purinergic receptors obtained by RNA-seq on primary cultures of human airway epithelial cells in 6 individuals [25].

**Figure 5 ijms-24-16160-f005:**
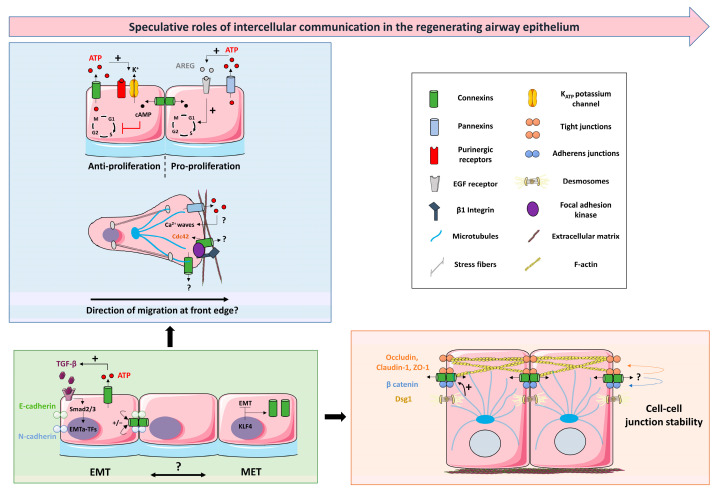
Hypothetical integrative scenario on the contribution of intercellular communication to airway epithelial regeneration. The blue box represents how connexins and pannexins, upon injury, could regulate proliferative and migratory signals through their channel-dependent and/or –inde-pendent function. The green box highlights the interplay between connexins and epithelial/mesenchymal markers in fine-tuning EMT/MET during the repair process. The orange box describes how connexins may ultimately be part of the barrier-stabilizing network through their physical and functional interaction with tight junctions, adherens junctions, and cytoskeleton components. + means positive action. ?: points to mechanisms requiring investigation. See text for details and abbreviations. The figure was created using pictures from Servier Medical Art. Servier Medical Art by Servier is licensed under a Creative Commons Attribution 3.0 Unported License (https://creativecommons.org/licenses/by/3.0/), accessed on 12 October 2023.

**Table 1 ijms-24-16160-t001:** Gene and protein nomenclature of connexins and pannexins subfamilies.

Gene	Gene Symbol	Protein Symbol
Gap Junction Alpha (GJA)	*GJA1*	Cx43
	*GJA3*	Cx46
	*GJA4*	Cx37
	*GJA5*	Cx40
	*GJA8*	Cx50
	*GJA9*	Cx59
	*GJA10*	Cx62
Gap Junction Beta (GJB)	*GJB1*	Cx32
	*GJB2*	Cx26
	*GJB3*	Cx31
	*GJB4*	Cx30.3
	*GJB5*	Cx31.1
	*GJB6*	Cx30
	*GJB7*	Cx25
Gap Junction Gamma (GJC)	*GJC1*	Cx45
	*GJC2*	Cx47
	*GJC3*	Cx30.2
Gap Junction Delta (GJD)	*GJD2*	Cx36
	*GJD3*	Cx31.9
	*GJD4*	Cx40.1
Gap Junction Epsilon (GJE)	*GJE1*	Cx23
Pannexin (PANX)	*PANX1*	Panx1
	*PANX2*	Panx2
	*PANX3*	Panx3

## Data Availability

Not applicable.

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
