# Peer review of "Intercellular Communication in Airway Epithelial Cell Regeneration: Potential Roles of Connexins and Pannexins"

_ijms, 2023, doi:10.3390/ijms242216160_

Round 1

Reviewer 1 Report

Comments and Suggestions for Authors

The review “Intercellular communication in airway epithelial cell regeneration” addresses the role of connexins and pannexins in maintaining the airway epithelium integrity and promoting tissue regeneration. However, a large part of the review is focused on other models, of epithelial and non-epithelial origin, which can be misleading. And it does not, strictly speaking, match the proposed scope. The current structure of the text does not separate the data on different models well and the text does not address the relevance of the cited works to the airway epithelium. Another important issue is the source of some illustrations (such as heatmaps in Figures 1-3) that is not discussed in the text. The review now has confusing parts and is not suitable for publication in its current form.

Major points:

1.       The origin of figures is not clear.

2.       Large passages of text are addressing cell models other than airway epithelium. The relevance of these models to the original topic is very moderately discussed.

3.       The structure of the review does not allow to separate cell models and the data obtained on them.

4.       About 25% references cite other published reviews and not experimental works.

5.       A number of passages lack references.

In text:

1.       Introduction, Lines 29-30 “Basal cells function as pluripotent 28 stem cells, capable of recapitulating the ciliated epithelium when exposed to an air–liquid 29 interface (ALI)” – lack reference.

2.       Connexin/pannexin expression in the airway epithelium, Lines 73-75 lack reference. Lines 86-88 lack reference.

3.       Connexin/pannexin expression in the airway epithelium, Lines 93-103 address Figure 1. However, it is not clear where the heatmaps come from. Is it a cited article or the re-analysis of the available RNAseq data made by the authors for the review? In the latter case a reference to data depository has to be made and the pipeline for analysis has to be provided. The legend to Figure 1 (Lines 119-123) does not provide this information either. Also, it is not clear what is depicted in Figure 1C – proteins or mRNAs? The data source of the picture is not clear either.

4.       Roles of intercellular communication in the well-polarized airway epithelium, Lines 134-135 lack reference. Line 142 – needs an experimental paper as a reference. Lines 143-144 lack reference. Line 174-175 – the relevance of this passage to the previous text is not clear.

5.       Speculative roles of intercellular communication in the regenerating airway epithelium, Lines 179-181 lack reference. Lines 188-190 lack reference. Lines 192-196 lack references.

6.       Speculative roles of intercellular communication in the regenerating airway epithelium, Lines 201-211 address Figure 2. Here it seems that the authors cite their previously published work. However, the source of the heatmaps is not explicitly stated. The figure legend (Lines 217-220)  does not provide this information either.­­­

7.       Connexins and proliferation, Line 230 – a typo in the author’s name. Line 233 – “The effect of Cx26 upregulation on the negative regulation of basal cells’ proliferation” is a bit of hard reading. Line 255 – the sense of the sentence is lost to me. Lines 256-260 lack references.

8.       Connexins, adhesion and migration – the subchapter (Lines 262-305) does not contain any references to airway epithelium models. The non-channel functions of connexins are first mentioned here and very moderately discussed.

9.       Subchapters “Connexins and epithelial integrity” (Lines 306-333) and “Connexins, differentiation and EMT” (Lines 334-374) cite many models other than airway epithelium without addressing their relevance. Lines 372-374 – the sense of the sentence is lost to me.

10.   Speculative roles of pannexins in the regenerating airway epithelium, Lines 405-406 and 415-416 address Figures 2 and 3. As before, the source of the heatmaps is not clear. Legend for Figure 3 (Lines 231-233) does not provide sufficient information.

11.   Speculative roles of pannexins in the regenerating airway epithelium, Lines 409-410 lack reference. Line 421 – the sentence is not clear.

12.   Concluding remarks. Figure 4 is very hypothetical. Considering the number of cell models discussed in the review and no specific summary on the airway epithelium provided anywhere in the text it is confusing.

13.   Concluding remarks. Lines 460-468 compare the canonical channel-dependent function of connexins to the channel-independent function, very briefly mentioned in the text before, and for the first time mention the protein modifications that may actually decide the intracellular fate of connexins and pannexins.  Isn’t it a bit late to first mention such things in the concluding remarks?

Comments on the Quality of English Language

Some editing of the manuscript is required.

Reviewer 2 Report

Comments and Suggestions for Authors

Badaoui & Chanson have gone through recent studies about the roles of connexins and pannexins in airway epithelial cell pathophysiology. 

1. connexins and pannexins should be included in the title of the article.

2. Lines 37-39, please cite the corresponding article to this.

3. Lines 49-51, how does releasing nucleotides relate to intercellular communication?

4. Line 86-89, please cite the corresponding article related to CFTR activity. 

5. For Fig 1C, I think you should include a heat map for individual cell populations here rather than showing transcriptomic differences by font size.

6. Lines 105-117, in vivo data and in vitro data should be clarified in this paragraph.

7. Lines 128-130, please cite the corresponding article about calcium waves and intercellular communication.

8. Line 174-176, Aren't mouse embryonic stem cells too far away from what you discuss in this article?

9. Lines 179-191, please describe the renewal process in a more precise and with proper references, the beginning of this paragraph has no reference to support your description. 

10. Lines 194-197, might need to rephrase the sentence to clarify the connection between cell junctions and regulation of molecules. 

11. Lines 233-234, from which study Cx26 is identified under the regulation of KLF4? Also please cite the corresponding article introducing the background of KLF4. 

12. I don't think references 46 and 47 are related to what you described in Lines 244-246.

13. Lines 355-357, is it accurate to compare cell behavior between adult stem cells and tumorigenic cells to investigate the underlying mechanisms? 

14. Fig 3A, what's the reason you use mRNA extracted from primary culture rather than tissue?  

Comments on the Quality of English Language

It would be necessary for this manuscript to go through professional academic proofreading to improve the overall quality. 

Round 2

Reviewer 2 Report

Comments and Suggestions for Authors

The structure and flow of the review article have been improved a lot compared to the last version. 

Comments on the Quality of English Language

The connection between paragraphs and contents is better.